# Health economic evaluations of preventative care for perinatal anxiety and associated disorders: a rapid review

Kalpa Pisavadia [ID],[1] Llinos Haf Spencer [ID],[1] Lorna Tuersley,[1] Rose Coates,[2] Susan Ayers,[2] Rhiannon Tudor Edwards [ID] [1]

[1]Centre for Health Economics and Medicines Evaluation, Bangor University, Bangor, UK
[2]City University of London, London, UK

**Correspondence to**
Kalpa Pisavadia;
Kalpa.pisavadia@bangor.ac.uk

## ABSTRACT

**Objectives** Perinatal mental health problems affect one in five women and cost the UK £8.1 billion for every year of births, with 72% of this cost due to the long-term impact on the child. We conducted a rapid review of health economic evaluations of preventative care for perinatal anxiety and associated disorders.

**Design** This study adopted a rapid review approach, using principles of the standard systematic review process to generate quality evidence. This methodology features a systematic database search, Preferred Reporting Items for Systematic Reviews and Meta-Analyses diagram, screening of evidence, data extraction, critical appraisal and narrative synthesis.

**Data sources** PubMed, Cumulative Index to Nursing and Allied Health Literature, Cochrane Library, Applied Social Sciences Index and Abstracts, PsycINFO and MEDLINE.

**Eligibility criteria for selecting studies** Studies that evaluated the costs and cost-effectiveness of preventative care for perinatal anxiety and associated disorders carried out within the National Health Service and similar healthcare systems.

**Data extraction and synthesis** A minimum of two independent reviewers used standardised methods to search, screen, critically appraise and synthesise included studies.

**Results** The results indicate a lack of economic evaluation specifically for perinatal anxiety, with most studies focusing on postnatal depression (PND). Interventions to prevent postnatal mental health problems are cost-effective. Modelling studies have also been conducted, which suggest that treating PND with counselling would be cost-effective.

**Conclusion** The costs of not intervening in maternal mental health outweigh the costs of preventative interventions. Preventative measures such as screening and counselling for maternal mental health are shown to be cost-effective interventions to improve outcomes for women and children.

**PROSPERO registration number** CRD42022347859.

## INTRODUCTION

The perinatal period refers to pregnancy and the first 12 months after childbirth.[1] One in five women experience mental health problems during this time, and the cost is estimated to be £8.1 billion for every year of

## STRENGTHS AND LIMITATIONS OF THIS STUDY

⇒ The strength of this rapid review is that it has highlighted costs associated with perinatal mental health interventions in a rigorous, novel way and has identified several gaps for future research.
⇒ The absence of health economic studies describing the range of public sector costs and costs to individuals from Scotland and Wales in relation to perinatal anxiety is a limitation of this rapid review.
⇒ Although health economic studies are showing the benefits of investing in postnatal depression, there are no published UK-based randomised controlled trials investigating perinatal mental health interventions, which include information on costs, which is a limitation of this rapid review.

births in the UK[2] (see online supplemental file 1 for a list of abbreviations). Maternal mental health problems include postnatal depression (PND) (also known as postpartum depression (PPD) internationally), characterised by depressed mood and anxiety, feelings of inadequacy and impaired infant bonding.[3]

Untreated maternal mental illness not only impacts mothers, but also adversely impacts their children, significantly contributing to wider societal and National Health Service (NHS) costs. Of the total costs of perinatal mental health difficulties in the UK, 72% is due to the long-term impact on the child.[2] Decreased maternal and infant bonding, reduced breastfeeding initiation rates and duration, low birth weight and poorer child growth have been associated with PND.[4] Children of mothers experiencing subclinical and persistently high depressive symptoms were twice as likely to have emotional and behavioural difficulties than children of mothers reporting minimal symptoms.[5] Delayed or impaired cognitive, linguistic, physical and psychological health development has been reported in infants and children with mothers with PND.[4] There is also a risk of intergenerational transmission of

socioeconomic disadvantage in which maternal mental illness impacts the child's quality of life by having a long-term adverse effect on education and employment prospects.[6][7] Public sector costs are likely to be significantly reduced by using a prevention strategy to reduce the incidence of poor maternal mental health.[7]

Despite the long-term risks of untreated maternal mental health issues, as of 2014, only 30–50% of women with perinatal mental health (PMH) problems were identified, and only 7% were referred to specialist care in the UK.[2] Most women with PMH problems did not access care.[2] This may have been particularly the case for women with mild to moderate PMH problems or less commonly recognised problems, such as anxiety, obsessive-compulsive disorder (OCD) or post-traumatic stress disorder (PTSD).[2] Access to care may also be limited by maternal time constraints and fears of being judged.[8]

The National Institute for Health and Care Excellence (NICE) recommends postnatal care for up to 8 weeks after birth.[9] Since 2015, it has been recommended that UK midwives carry out emotional well-being checks at antenatal check-ups and at each postnatal contact up to 8 weeks after birth. In 2018, the National Collaborating Centre for Mental Health worked with NICE to develop the Perinatal Mental Health Care Pathway.[10] The guidance in that report follows a process agreed upon by NICE and sets out

pathways to deliver a strategic transformation of PMH care. Psychological interventions, either alone or in conjunction with pharmacological treatment, are recommended for complex or severe mental health problems following referral to a specialist community perinatal mental health team.[1]

Since 2015, there have been improvements to funding plans and commitments in the provision of more specialist Community Perinatal Mental Health Services across the UK. For example, in 2019, the Scottish government revealed that £52 million would be spent on improving access to perinatal and infant mental health services, and from 2018 to 2020, the Welsh government increased recurrent annual funding from £1.5 million to £2.5 million for specialist PMH services.[11] In England, the government committed £365 million to provide specialist perinatal community services across the country, as announced by NHS England in April 2019.[9] It is imperative that proactive planning and cost-effective preventative solutions are a public policy priority.[6]

## Aim

This review aims to investigate the type of health economic evaluations of preventative care for perinatal anxiety and associated disorders carried out within the NHS and similar healthcare systems.

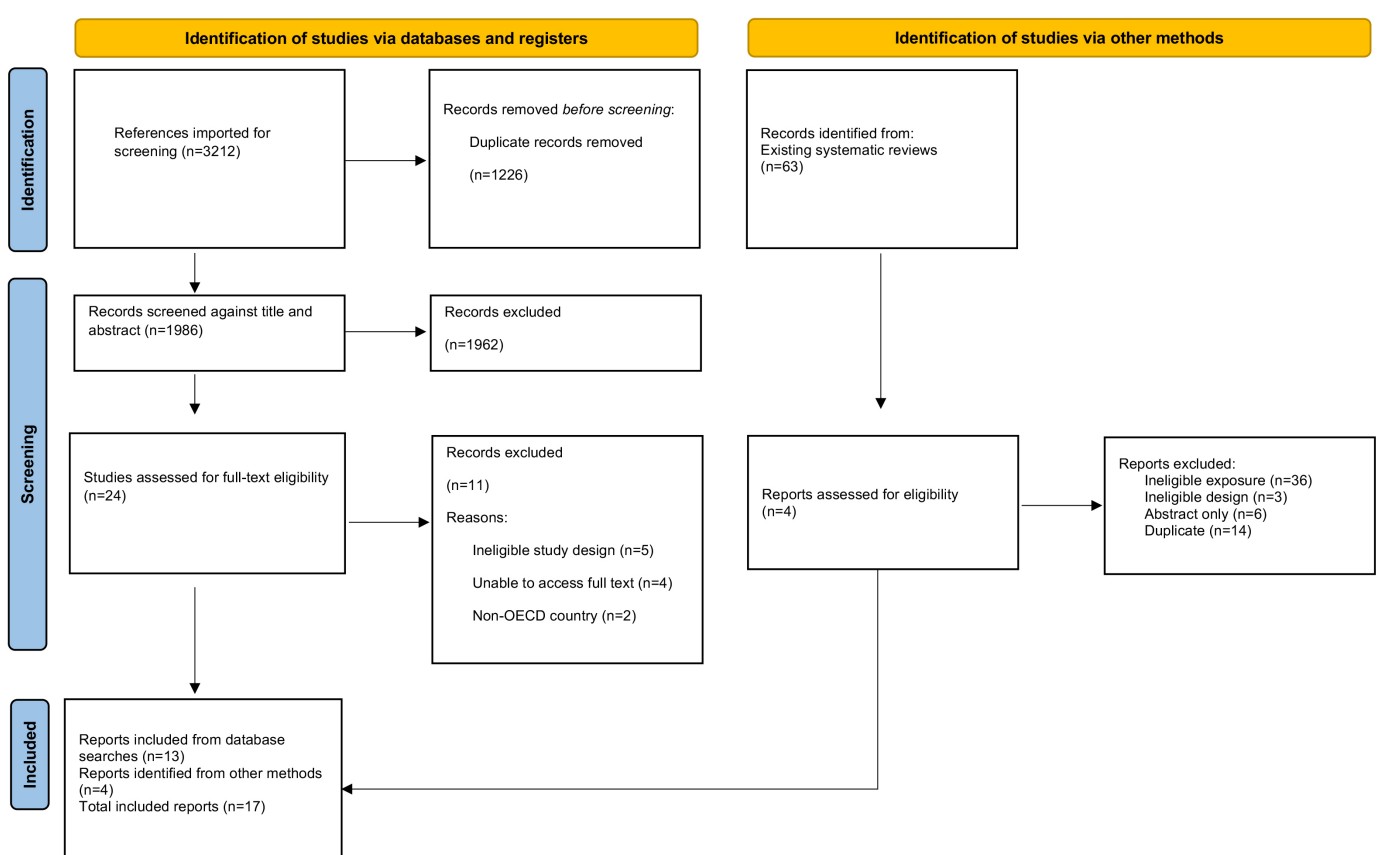

**Figure 1** Preferred Reporting Items for Systematic Reviews and Meta-Analyses study selection flow chart (Page *et al*).[12] OECD, Organisation for Economic Co-operation and Development.

**Table 1** Participants, Intervention/Exposure, Comparator and Outcomes framework

| Question | |
|---|---|
| What is the cost of care for women experiencing perinatal anxiety and associated disorders? | |
| Participants | Pregnant women or perinatal women |
| Intervention/ exposure | Perinatal anxiety and associated disorders |
| Comparator | No comparator |
| Outcomes | Costs of primary care and support services for women experiencing perinatal anxiety and associated disorders |
| Study considerations | |
| Primary research, secondary research, grey literature and preprints | |
| Databases | |
| PubMed, CINAHL, Cochrane Library, ASSIA, PsycINFO and MEDLINE | |

## METHODS

This review used principles from the standard systematic review process to generate quality evidence in a shorter time frame. This methodology included a systematic database search, Preferred Reporting Items for Systematic Reviews and Meta-Analyses diagram[12] (see figure 1) screening of evidence, data extraction, critical appraisal and narrative synthesis. This revised methodology is used by the Health and Care Research Wales Evidence Centre.[13–15] Cost-effectiveness outcomes are reported according to the Professional Society for Health Economics and Outcomes Research guidelines.[16]

### Patient and public involvement

None.

### Search strategy

The key evidence sources included PubMed, Cumulative Index to Nursing and Allied Health Literature, Cochrane Library, Applied Social Sciences Index and Abstracts, PsycINFO and MEDLINE. The search terms consisted of words related to perinatal anxiety and/or depression, health and psychiatric services and economic evaluation terms. The searches were conducted on 23 April 2022. Mendeley reference management software was used to manage study articles found and remove duplicates. See online supplemental file 1 for the full search strategy.

The eligibility criteria for the review are presented in table 1 and are based on the Population, Intervention, Comparison and Outcome framework.[17] This consisted of peer-reviewed economic evaluations of perinatal anxiety and associated disorders such as PND and PTSD from Organisation for Economic Co-operation and Development (OECD) countries in English published after January 2000.

### Selection of studies

One reviewer (KP) independently selected potentially eligible studies based on a screening of titles and abstracts. Two reviewers (LHS and KP) selected additional studies from existing systematic reviews. The full texts of selected studies were assessed for eligibility by three reviewers (KP and LHS, with mediation by LT) in the data extraction process.

### Data extraction

Data extraction and study quality assessment were performed by three reviewers (KP, LHS, LT). Data were collected on country, study design, intervention type, data collection methods and dates, sample size and type of participants (see online supplemental file 2 for data extraction tables).

### Quality assessment

The quality assessment was undertaken by two reviewers (LHS and KP), and four papers were checked by a third reviewer for quality assurance purposes (LT). The Drummond checklist[18] was used for the quality appraisal of health economic papers, and the *CH*ecklist for critical *A*ppraisal and data extraction for systematic *R*eviews of prediction *M*odelling *S*tudies (CHARMS) checklist was used for the modelling studies.[19] The Joanna Briggs Institute critical appraisal tools were used for the quality appraisal, randomised clinical trials, cohort studies and cross-sectional studies[20–22] (see online supplemental file 1).

## RESULTS

Searches of databases yielded 3212 results, of which 1226 duplicates were removed. The remaining 1986 results were screened against titles and abstracts, and an additional four papers were retrieved from existing systematic reviews. A total of 17 papers met the criteria for full-text screening. Eleven papers were excluded due to not being able to access the full text (n=4), ineligible study design (n=5) or lack of relevancy (non-OECD country) (n=2). Seventeen studies were included in this rapid review (see figure 1 and table 2).

Of these 17 included papers, there were cost-effectiveness studies (n=5), modelling studies (n=6), cost–benefit study (n=1), a cost-analysis study (n=1) and cost of illness studies (n=4). All included studies were peer reviewed. The included studies were categorised according to main intervention: children, prevention, cost of maternal health, cost of single interventions and comparison cost of interventions. The following discussion provides a more detailed overview of the findings.

The included papers are organised under three different themes: perinatal anxiety, perinatal depression and perinatal health and well-being. These studies are detailed below, and all non-UK prices have been converted to pound sterling currency and inflated to the

**Table 2** Map of maternal cost of illness studies by evidence type (including studies on depression, anxiety and maternal health and well-being)

| Type of evidence | Type of intervention | | | | | Number of studies |
| | Children | Prevention | Cost of maternal health | Cost of single interventions | Comparison cost of interventions | |
|---|---|---|---|---|---|---|
| Cost-effectiveness | | Petrou et al[3] | | Morrell et al[44] | Henderson et al[43] | 5 |
| | | Ride et al[28] | | Stevenson et al[30] | | |
| Cost–benefit | | | | | Grote et al[31] | 1 |
| Cost-analysis | Moore Simas et al[4] | | | | | 1 |
| Cost of illness | | | Petrou et al[32] | | | 4 |
| | | | Dagher et al[33] | | | |
| | | | Ammerman et al[34] | | | |
| | | | Roberts et al[35] | | | |
| Economic modelling studies | Bauer et al[6] | Counts et al[36] | Franta et al[37] | | | 6 |
| | Ride[45] | Wilkinson et al[38] | Chojenta et al[46] | | | |
| Total number of studies | 3 | 4 | 6 | 2 | 2 | 17 |

latest available prices (□□ denotes inflation and conversion, □ denotes inflation only, ∧ denotes conversion to GBP only .[23–27]

### Summary of studies including perinatal anxiety

This review found one economic evaluation focusing on perinatal anxiety.[28 29] This study consisted of a cost-effectiveness, cost-utility analysis and cluster randomised controlled trial (RCT) of the What Were We Thinking (WWWT) intervention.[28] WWWT is a psychoeducational intervention targeted at the partner relationship, management of infant behaviour and parental fatigue for the prevention of postnatal maternal mental health problems (see table 3 for further details). There were no statistically significant differences in either costs or effectiveness.

### Summary of studies including perinatal depression

Fifteen studies focused on perinatal depression.[3 4 6 30–41] A cross-sectional study from the USA conducted between 2006 and 2011 investigated the out-of-pocket expenses and insurer expenses of depressed mothers compared with non-depressed mothers.[34] Depressed mothers were more likely to incur insurer out-of-pocket expenses (£1285 vs £853□□) and have higher insurer expenses (£10 485 vs £7508□□).

One study used the perspective of the public sector, individuals and society to examine some of the outcomes and long-term economic implications experienced by offspring who have been exposed to perinatal depression in a South London cohort.[6] Bauer et al[6] found that for each child exposed to perinatal depression, public sector costs exceeded £3380□, costs due to reduced earnings were £1562□ and health-related quality of life loss was valued at £3760□.

A decision analytical model used a simulated cohort of 1000 Medicaid-enrolled pregnant individuals to evaluate the healthcare costs for individuals receiving PND preventive intervention or not, for 1–5 years post partum.[36] This study found that providing preventive interventions for PPD resulted in an estimated 5-year saving of £602□□.

A cross-sectional study in the USA investigated expenditure on healthcare services from hospital discharge until 11 weeks post partum.[33] There was a significant difference in healthcare expenditure between depressed and non-depressed women. The Edinburgh Postnatal Depression Scale (EPDS) was used to measure depression.[42] The total cost of all mental health counselling visits for the depressed group (n=31) was £165□□, and the cost for the non-depressed group (n=607) was £15.50□□ (in 2007). This was a statistically significant difference (p<0.001).

Using a theoretical cohort of 180 000 individuals, a decision analytical model compared outcomes in pregnant adolescents who received counselling interventions versus those who did not.[37] This study found that it is cost-effective to refer all pregnant adolescents for preventive counselling interventions. Within the theoretical cohort for counselling, there were 8935 fewer cases of PND, 1606 fewer cases of chronic depression, 166 fewer preterm deliveries, 4 fewer neonatal deaths, 20 fewer cases of sudden infant death syndrome and 1 fewer case of cerebral palsy. In total, there were 21 976 additional quality-adjusted life years (QALYs) and cost-savings of £183 463 169, ∧making it the dominant strategy that had better outcomes with lower costs.

An RCT compared a multicomponent collaborative care intervention for depression (MOMCare—a choice of brief interpersonal psychotherapy or pharmacotherapy

**Table 3** Methodological considerations and cost-effectiveness results

| Lead author (year) | Intervention | Perspective (reasons) | Time horizon used in economic evaluation (reasons) | Discounting | Key cost-effectiveness results |
|---|---|---|---|---|---|
| Henderson[43] (2019) | **Intervention group:** PoNDER: health visitor (HV) training to assess postnatal depression (PND) and deliver psychological approaches to women at risk of depression **Control group:** usual care | NHS and social care perspective | Resource use data from 6 weeks to 6 months were collected on a resource use log completed by HVs based on their own and GP records. | No discounting was necessary due to the duration of the follow-up period. | Costs and outcomes data were available for 1459 participants. 6-month adjusted costs were £82 lower in intervention than control groups, with 0.002 additional QALY gained. The probability of cost-effectiveness at £20 000 was very high (99%). |
| Morrell[44] (2000) | **Intervention group:** up to 10 home visits in the first postnatal month of up to 3 hours' duration by a community postnatal support worker **Control group:** usual care | NHS perspective | Up to 10 home visits in the first postnatal month of up to 3 hours' duration by a community postnatal support worker and a 6-month follow-up | No | Cost data showed that at 6 weeks, the mean total NHS costs were £635 for the intervention group and £456 for the control group (p=0.001). At 6 months, figures were £815 and £639 (p=0.001). However, due to there being no differences between the groups in use of social services or personal costs, no cost-effectiveness analysis was conducted. |
| Petrou[3] (2006) | **Intervention group:** counselling and specific support for the mother relationship, targeted at women at high risk of developing PND **Control group:** usual care | The economic evaluation was conducted from a public sector perspective. | The time horizon for the economic evaluation mirrored the time horizon for the randomised controlled trial, namely the period between randomisation and 18 months post partum. | Various discounting rates were applied as necessary: 0%, 1.5%, 3%, 6% and 10%. | The mean health and social care costs were estimated at £2396.9 per mother–infant dyad in the preventive intervention group and £2277.5 per mother–infant dyad in the routine primary care group, providing a mean cost difference of £119.5 (bootstrap 95% CI −535.4, 784.9). At a willingness-to-pay threshold of £1000 per month of PND avoided, the probability that the preventive intervention is cost-effective is 0.71 and the mean net benefit is £383.4 (bootstrap 95% CI −£863.3, £1581.5). |
| Ride[28] (2016) | **Intervention group:** What Were We Thinking (WWWT)—a psychoeducational intervention targeted at the partner relationship, management of infant behaviour and parental fatigue **Control group:** usual care | A range of perspectives including patient, NHS and social services | The time horizon of 6 months mirrored the trial follow-up period. No | No discounting was necessary due to the duration of the follow-up period. | The incremental cost-effectiveness ratios were $A36 451 per QALY gained and $A152 per percentage point reduction in 30-day prevalence of depression, anxiety and adjustment disorders. The estimate lies under the unofficial cost-effectiveness threshold of $A55 000 per QALY; however, there was considerable uncertainty surrounding the results, with a 55% probability that WWWT would be considered cost-effective at that threshold. |
| Stevenson[30] (2010) | **Intervention group:** group cognitive-behavioural therapy (gCBT) **Control group:** usual care | Health sector perspective | Treatment up to 8 weeks and a 6-month follow-up | No discounting was necessary due to the duration of the follow-up period. | The use of gCBT does not appear to be cost-effective. The mean cost per QALY from the stochastic analysis was estimated to be £36 062; however, there was considerable uncertainty around this value. The expected value of perfect information was estimated to be greater than £64 million; the key uncertainties were in the cost per woman of providing treatment and in the statistical relationship between changes in the Edinburgh Postnatal Depression Scale values and changes in the Short Form–6 Dimensions values. The expected value of perfect partial information for both of these parameters was in excess of £25 million. |

GP, general practitioner; NHS, National Health Service; QALY, quality-adjusted life year.

or both) with enhanced maternity support services (MSS-Plus) in the public health system of Seattle, USA.[31] The incremental benefit and cost and the net benefit for women with major depression and PTSD were estimated. When controlled for baseline depression severity, women with probable depression and PTSD in MOMCare had 68 more depression-free days over 18 months than those in MSS-Plus (p<0.05). There was an additional £1943□□ depression care cost per MOMCare participant with comorbid PTSD. The incremental net benefit of MOMCare was positive if depression-free days were valued below £18□□. For women with probable major depression and PTSD, MOMCare had a significant clinical benefit over MSS-Plus, with only a moderate increase in health services cost.

A cluster RCT of health visitors trained to assess PND and deliver psychological approaches to women at risk of depression plus either a cognitive–behavioural approach or a person-centred approach weekly for 8 weeks was conducted in 2019.[43] A cost-effectiveness analysis was run parallel to this for all mothers at low risk of depression in accordance with the EPDS at 6 months postnatally. This study found that cognitive–behavioural therapy (CBT) had a marginally higher probability of being cost-effective than a person-centred approach. The short time horizon of 6 months postnatally means that the risks of long-term adverse effects were not factored into the analysis.

A cohort study with a sample size of 135 678 mother–child pairs with and without PND exposure revealed similar findings.[4] The results of this analysis suggest that the health resource utilisation and costs over the first 24 months of life in children of mothers with PND exceeded that of children of mothers without evidence of PND (£22 940□□ and £20 487□□, respectively). This was a significant difference of £2453.

A longitudinal study (18 months) conducted in 2002 estimated the economic costs of PND in a cohort of women at high risk of developing the condition with the use of an RCT to identify women considered to be of high risk.[32] Unit costs were applied to estimates of health and social care resource use made by 206 women and their infants recruited from antenatal clinics, and net costs per mother–infant dyad over the first 18 months post partum were estimated. This study found that costs were £587□ higher for women with PND than for women without PND. Economic costs were particularly higher for women with extended experiences of the condition.

A cost-effectiveness analysis of preventive interventions, consisting of counselling and support for the mother–infant relationship at high risk of developing PND, was conducted in 2006.[3] This study found that given the negative impact of PND on later child development, preventive interventions are likely to be cost-effective even at relatively low willingness-to-pay thresholds for preventing 1 month of PND during the first 18 months post partum. The mean health and social care costs were estimated at £3345□ per mother–infant dyad in the preventive intervention group and £3277□ per mother–infant dyad in the

routine primary care group, providing a mean cost difference of £166□.

A cross-sectional study of 1250 mothers of infants in a Canadian setting used the EPDS to investigate the costs associated with perinatal depression.[35] It was found that costs were notably different for mothers with and without depression. The total cost for health and social care was £833□□ for mothers with depression and their infants, compared with £406□□ for those with lower depression scores. This was statistically a significant difference at p<0.01.

An economic evaluation conducted in 2010 compared the cost-effectiveness of group CBT (gCBT) compared with routine primary care for women with PND in the UK.[30] This economic evaluation found that gCBT does not appear to be cost-effective due to the lack of literature providing robust information. Only one study, an RCT, was deemed applicable to the decision problem.

A cost-effectiveness analysis found that screening for and treating PPD is a cost-effective intervention.[38] This study followed a hypothetical cohort of 1000 pregnant women experiencing one live birth over a 2-year time horizon. The analysis found that screening for and treating PND and psychosis produced 29 more healthy women at the cost of £938□□ per woman. The incremental cost-effectiveness ratios (ICERs) of the intervention branch compared with usual care were £13 702□□ per QALY gained (below the commonly accepted willingness-to-pay threshold of $50 000 per QALY gained) and $10 182 per remission achieved.

### Summary of studies including maternal health and well-being

This review found four studies relating to the health and well-being of perinatal women.[44–47] An RCT conducted in 2000 aimed to establish the relative cost-effectiveness of postnatal support in the community in addition to the usual care provided by the community midwives.[44] Three hundred and eleven women were allocated to the intervention of up to 10 home visits by a community postnatal support worker. No health benefit was found for additional home visits by community postnatal support workers compared with traditional community midwifery visiting, as measured by the Short Form-36 (SF-36) measure. At 6 months, there was no significant improvement in health status among the women in the intervention group despite there being a significant difference in costs of £1250□ (intervention group) and £980□ (usual care group) (p=0.001). Although there were no savings to the NHS over 6 months after the introduction of the community postnatal support worker service, the women in the intervention group were very satisfied with the support worker visits.

Authors have suggested that prenatal interventions that do not seem cost-effective in the short term may be cost-effective over a longer time horizon.[45] A decision analytical modelling study noted that it is important to consider caregiving and family health effects in the outcomes of maternal health studies.[45] By not including broader sets

of costs and outcomes, resources in postnatal mental health may be misallocated. As a result, some women may not benefit as much from interventions that might be cost-effective given a broader time horizon. The uncertainty surrounding the results in the decision analytical model may reflect decisions and investment in PND interventions.

A modelling study from Australia, published in 2019, used cohort data from 1921 to 1995 and found that the healthcare costs for postnatal women who had poor mental health prior to birth were £1066∧.[46] This is, on average, 11% more than for mothers with no history of poor mental health. These figures do not include out-of-pocket expenditure for the women who may have also purchased their own over-the-counter medications and had other patient expenses which were not captured in the analysis.

## DISCUSSION

The aim of this review was to investigate the type of health economic evaluations of preventative care for perinatal anxiety and associated disorders carried out within the NHS and similar healthcare systems. Seventeen papers were included in this review from Australia, Canada, Ireland, the USA and the UK, each examining maternal mental health.

The results indicate a lack of economic evaluation specifically for perinatal anxiety, with most study articles focusing on PND.[30] Only two included papers focused on anxiety, with one being a systematic review looking at anxiety alongside depression.[29] The other was an economic evaluation of a maternal mental health intervention. Treatments for maternal mental health in the WWWT intervention consisted of health visitors with psychiatric training and group sessions focusing on parenting confidence and emotional well-being with online and face-to-face components.[28] The WWWT intervention shows promise as a preventive intervention, but uncertainty surrounding cost-effectiveness. The analysis showed no statistically significant difference in costs or outcomes between the intervention and control groups, with the intervention estimated to cost £74.48 per participant.

Most of the studies included (n=15 of the 17 included studies) focused on the cost of services and interventions for PND. The evidence suggests significant health resource costs outside of mental health services as well as social care costs for PND for mother and mother–infant dyad. Costs were significantly higher for children of mothers with PND than for children of mothers without PND. This was a statistically significant difference of £2453 (p<0.001).[4]

Counselling was found to be a cost-effective, preventative intervention for pregnant adolescents.[37] Using a hypothetical cohort, one study found that counselling was a cost-effective preventative measure, leading to fewer cases of perinatal and chronic depression.[37]

Another study estimated that group counselling (costing £114 per mother) cost around £73□ less than individual counselling (£187 per mother) for mothers with PND.[36] This study found that screening for PND costs less than £2 per mother.[36] Studies that combined screening for PND with an intervention were also found to be cost-effective, resulting in 29 more healthy women at a cost of £938□□ per woman.[38] The ICERs of the intervention branch compared with usual care were $13 857 per QALY gained (below the commonly accepted willingness-to-pay threshold of $50 000 per QALY gained) and $10 182 per remission achieved.

Within this review, the EPDS, a validated measure for PND and anxiety,[42] was the most frequently used instrument to detect perinatal and PND in the included studies, followed by the SF-36 scale, postal questionnaires such as the Ontario health survey, Health and Social Service Utilisation Questionnaire, blinded telephone assessments and medical records, Medicaid data, resource use logs completed by health visitors based on general practitioner records, and prospective diaries and face-to-face interviews.

In summary, screening was found to be a relatively low-cost method of identifying women in need of mental health support during the perinatal period. Interventions to prevent postnatal mental health problems were found to be cost-effective.[28] Also, two modelling studies found that treating PND with counselling would be cost-effective.[30 38]

Future research in this area should investigate how best to screen all mothers to prevent and treat further adverse outcomes such as anxiety, OCD or PTSD.[2] Various psychosocial methods could be used to screen and provide treatment over the telephone, online or face-to-face. Interventions could be provided by a range of healthcare professionals, such as midwives, health visitors, counsellors, psychologists and psychiatrists. The effectiveness and cost-effectiveness of each intervention, including screening, should be evaluated.

Web-based approaches are already promising to be cost-effective solutions to support mothers in the perinatal period. A recent cost-effectiveness study alongside an RCT in Singapore evaluated a web-based approach for delivering a psychoeducational intervention.[14] This web-based approach was cost-effective in supporting first-time mothers and provided the best improvements in self-efficacy, social support and psychological well-being of mothers in the perinatal period. Most women of childbearing age, including women who reside in rural areas, now have access to the internet in the UK and similar healthcare systems. Being able to access support and treatment using online resources has widened access to care to postnatal care support.

### Limitations of this study

Although this study conducted a thorough systematic search, only peer-reviewed literature was included. Relevant grey literature in this area may provide more insights

into preventative interventions for maternal mental health that could be cost-effective. The findings in this area are limited by the literature available, particularly the absence of published RCTs with cost data, which would provide a rigorous method of hypothesis testing of PMH interventions.

## CONCLUSION

This review demonstrated that very few economic evaluations have focused on perinatal anxiety, and those which reported on cost of perinatal depression had short time horizons which did not allow for long-term outcomes for the mother and child dyad to be addressed. However, there was some evidence that preventative measures, such as PND screening, combined with treatment, such as counselling for maternal mental health, are proven to be effective interventions to improve outcomes for women and children.

### Recommendations

It is recommended that:

▶ Mothers should be screened for maternal mental health issues to identify mothers at risk and provide treatment, leading to better outcomes for the mother and child dyad.

▶ Studies focusing on interventions for perinatal anxiety as a distinct condition to other mental health issues such as depression should be conducted.

▶ Cost of intervention studies related to perinatal anxiety should be conducted.

**Acknowledgements** We thank the wider MAP ALLIANCE team from City University of London and the University of Stirling for input into the development of this review. We would also like to thank Yasmine Noorani, Academic Support Librarian at Bangor University, for her assistance in creating our search strategy. Additional thanks to Dr Catherine Lawrence for early input and feedback on this paper.

**Contributors** The review was conceived by RTE, and the protocol was developed by LT, KP and LHS. KP acts as the guarantor of this paper. Searches were undertaken by KP. Article screening was carried out by KP and LHS with mediation by LT. Quality appraisal was undertaken by KP, LHS and LT. Data were interpreted by all authors. The manuscript was drafted by KP and LHS and critically reviewed by authors RC, SA and RTE.

**Funding** This review is to complement the MAP ALLIANCE Study, funded by the National Institute for Health and Care Research (NIHR) (award ID: NIHR133727). This rapid review was partially funded by Health and Care Economics Cymru (HCEC), an organisation funded by Health and Care Research Wales.

**Competing interests** None declared.

**Patient and public involvement** Patients and/or the public were not involved in the design, or conduct, or reporting, or dissemination plans of this research.

**Patient consent for publication** Not applicable.

**Ethics approval** This study involves human participants and the MAP ALLIANCE Study received ethical approval from the West of Scotland Research Ethics Committee on 6 May 2022 (REC 3; reference: 22/WS/0029). Participants gave informed consent to participate in the study before taking part.

**Provenance and peer review** Not commissioned; externally peer reviewed.

**Data availability statement** No data are available. No datasets were generated and/or analysed for this study.

**ORCID iDs**
Kalpa Pisavadia http://orcid.org/0000-0003-1435-163X
Llinos Haf Spencer http://orcid.org/0000-0002-7075-8015
Rhiannon Tudor Edwards http://orcid.org/0000-0003-4748-5730

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
