## [Reviewer comments · BMJ Open]

ARTICLE DETAILS

TITLE (PROVISIONAL)	Health economic evaluations of preventative care for perinatal anxiety and associated disorders: A rapid review
AUTHORS	Pisavadia, Kalpa; Spencer, Llinos; Tuersley, Lorna; Coates, Rose; Ayers, Susan; Edwards, Rhiannon

VERSION 1 – REVIEW

REVIEWER	Sumathipala, Athula Keele University School of Medicine, Research Institute for Primary Care & Health Sciences
REVIEW RETURNED	03-Nov-2022

GENERAL COMMENTS	Well written timely piece of work
-----------------------------------

REVIEWER	Laksham, Karthik Jawaharlal Institute of Postgraduate Medical Education and Research, Community Medicine
REVIEW RETURNED	16-Nov-2022

GENERAL COMMENTS	Dear Author, Thank you for submitting your valuable research. The research question is clearly described and the search strategy is reported sufficiently to reproduce. However, for this Review of Cost Effectiveness, the following points have to be addressed:- 1. The design of the study is not clear. Title mentions as Cost Effectiveness, however the research question and many of the included studies are of Cost-of-Illness studies. Table no.1 mentions as No Comparator. Cost effectiveness studies compares an intervention with a comparator.2. Limitations of this study have to be described.3. Details about Time horizon and discounting, Adjustment of inflation, Interventions compared, Method(s) for valuation of effectiveness and utility outcomes, Compliance/adherence with intervention, Health outcomes (eg QALYs), Uncertainty (eg, sensitivity analyses, subgroup analyses) have to be mentioned for individual studies. These can be presented as a table. Kindly refer to this attached article "Critical Appraisal of Systematic Reviews With Costs and Cost-Effectiveness Outcomes: An ISPOR Good Practices Task Force Report" for better reporting of Reviews with Cost effectiveness outcome
--

REVIEWER	Bauer, Annette London School of Economics and Political Science
REVIEW RETURNED	13-Feb-2023

GENERAL COMMENTS	Thank you for giving me the opportunity to read you rapid review. My concern with the submitted work is that aims, methods, results and discussion sections are not aligned. There are inconsistencies, for example, between the title, abstract and the main text as well as within the main text with regards to: whether the review was seeking to examine the costs of delivering interventions, the cost consequences linked to interventions, the cost-effectiveness of interventions or the costs linked to the condition. It also not always clear whether the focus was on perinatal anxiety, on depression and possibly other conditions. There are also other major issues, for example related to the use of economic evaluation terminology which at times is applied incorrectly (e.g. the use of the term cost of illness studies in Table 2 seems to have been confused with cost-effectiveness studies). However, the issue of inconsistency and unclarity is so grave that it does need to be addressed in my view before the work can be reviewed again.
---

VERSION 1 – AUTHOR RESPONSE

No.	Reviewer comment	Comment by MAP ALLIANCE team	Page no.
Reviewer one	Well written timely piece of work	Thank you for the positive feedback.	N/A
Reviewer two	1. The design of the study is not clear.	(Haby et al., 2016) has been added as a reference in the methodology section.	Page 5
	Title mentions as Cost Effectiveness, however the research question and many of the included studies are of Cost-of-Illness studies.	The title has been amended to Health economic evaluations of preventative care for perinatal anxiety and associated disorders: A rapid review	Page 1.
	Table no.1 mentions as No Comparator. Cost effectiveness studies compares an intervention with a comparator.	The new Table 3 describes the comparator in the cost-effectiveness studies.	New Table 3 is on Page 10.
	2. Limitations of this study have to be described.	Limitations of each paper have been	Pages 11-15.

		described in the main results text.	
	3. Details about: Time horizon and discounting,	The new Table 3 describes the time horizon in the cost-effectiveness studies.	New Table 3 is on Page 10.
	Adjustment of inflation, Interventions compared,	The new Table 3 describes the discounting method used in the cost-effectiveness studies.	New Table 3 is on Page 10.
	Method(s) for valuation of effectiveness and utility outcomes	More details regarding effectiveness and utility outcomes have been included to the results section.	P. 11-15.
	Compliance/adherence with intervention, Health outcomes (e.g. QALYs),	More details have been included to the results section.	P. 10-15.
	Uncertainty (e.g., sensitivity analyses, subgroup analyses) have to be mentioned for individual studies. These can be presented as a table.	The new Table 3 describes the uncertainty in the cost-effectiveness studies.	New Table 3 is on Page 10.
	Kindly refer to this attached article "Critical Appraisal of Systematic Reviews With Costs and Cost-Effectiveness Outcomes: An ISPOR Good Practices Task Force Report" for better reporting of Reviews with Cost effectiveness outcome	The Mandrik et al (2021) reference (Mandrik et al., 2021) has been added to the methodology section.	Page 5 (in the Methodology section).
Reviewer three	My concern with the submitted work is that aims, methods, results and discussion sections are not aligned.	The paper has been rewritten to ensure that the aims, methods, results and	Whole paper

		discussion are better aligned.	
	There are inconsistencies, for example, between the title, abstract and the main text as well as within the main text with regards to: whether the review was seeking to examine the costs of delivering interventions, the cost consequences linked to interventions, the cost-effectiveness of interventions or the costs linked to the condition.	The paper has been rewritten to ensure that the aims, methods, results and discussion are aligned	Whole paper
	It also not always clear whether the focus was on perinatal anxiety, on depression and possibly other conditions.	The focus was on perinatal anxiety, but we report other relevant evidence focusing on postnatal depression.	Whole paper
	There are also other major issues, for example related to the use of economic evaluation terminology which at times is applied incorrectly (e.g. the use of the term cost of illness studies in Table 2 seems to have been confused with cost-effectiveness studies).	The health economics terminology has now been reviewed and the included studies are now in their relevant column.	See Table 2 for included studies. Page 9.